# Assessing the Pyloric Caeca and Distal Gut Microbiota Correlation with Flesh Color in Atlantic Salmon (*Salmo salar* L., 1758)

**DOI:** 10.3390/microorganisms8081244

**Published:** 2020-08-16

**Authors:** Chan D. H. Nguyen, Gianluca Amoroso, Tomer Ventura, Abigail Elizur

**Affiliations:** 1Genecology Research Centre and School of Science and Engineering, University of the Sunshine Coast, Sippy Downs, Queensland 4556, Australia; cnguyen1@usc.edu.au (C.D.H.N.); gianluca.amoroso@petuna.com (G.A.); 2Petuna Aquaculture, East Devonport, Tasmania 7310, Australia

**Keywords:** Atlantic salmon, *Salmo salar*, microbiota, flesh color, banding, pigmentation

## Abstract

The Atlantic salmon (*Salmo salar* L., 1758) is a temperate fish species native to the northern Atlantic Ocean. The distinctive pink–red flesh color (i.e., pigmentation) significantly affects the market price. Flesh paleness leads to customer dissatisfaction, a loss of competitiveness, a drop in product value and, consequently, severe economic losses. This work extends our knowledge on salmonid carotenoid dynamics to include the interaction between the gut microbiota and flesh color. A significant association between the flesh color and abundance of specific bacterial communities in the gut microbiota suggests that color may be affected either by seeding resilient beneficial bacteria or by inhibiting the negative effect of pathogenic bacteria. We sampled 96 fish, which covered all phenotypes of flesh color, including the average color and the evenness of color of different areas of the fillet, at both the distal intestine and the pyloric caeca of each individual, followed by 16S rRNA sequencing at the V3-V4 region. The microbiota profiles of these two gut regions were significantly different; however, there was a consistency in the microbiota, which correlated with the flesh color. Moreover, the pyloric caeca microbiota also showed high correlation with the evenness of the flesh color (beta diversity index, PERMANOVA, *p* = 0.002). The results from the pyloric caeca indicate that *Carnobacterium*, a group belonging to the lactic acid bacteria, is strongly related to the flesh color and the evenness of the color between the flesh areas.

## 1. Introduction

The Atlantic salmon, *Salmo salar,* farmed in Tasmania, is one of the highest value aquaculture products in Australia and contributed AUD 756 million of the total AUD 1.3 billion of Australian aquaculture production in 2016–2017 (Australian Bureau of Agricultural and Resource Economics and Sciences, ABARES, 2018). The pink–red flesh color in Atlantic salmon is considered as the most important criterion for a high-quality and high-value product [1]. Atlantic salmon reared in Tasmania are an introduced species and, despite a certain degree of genetic adaptation over a few decades, are still challenged by living in the upper limit of their temperature tolerance during summer. Therefore, when the water temperature exceeds 20 °C for a prolonged period of time during summer in Tasmania, the temperate Atlantic salmon suffer from thermal stress, aggravated by poor feeding as extreme as starvation, which eventually leads to a loss in flesh color in part of the stock [2]. This loss in flesh color, when persisting up to the harvest stage, can lead to a reduction in product value and moderate economic losses [2,3].

In Atlantic salmon, there are two patterns of flesh color loss: general flesh paleness and localized color loss, which consists of differential color tones between regions of the fillet [2,3]. The color difference is usually between the front dorsal region, which is more prone to discoloration, and the central back region of the fillet, which retains color better [4]. This phenomenon, commercially known as banding, is annually discernible between February and August in Tasmania, and it is usually recovered in most of the stock by harvest time. During this time, the fluctuation of water temperature is one of the reasons causing stress in fish [5] and induces feed reduction or cessation. As a result, water temperature can also affect the fish gut microbiota in Atlantic salmon, as previously demonstrated [6,7,8] and confirmed in our recent study [9], where we showed that changes in the distal gut microbiota profile correlates with flesh color variation across different time points during the relevant period (February–August). Although it is clear that overall temperature fluctuation has an impact on color loss during that period [3], flesh color variation correlation with the gut microbiota is independent of the change in water temperature [9], since fish with reduced flesh color have similar microbiota profiles in the distal gut regardless of the time point tested. However, only the distal gut region, which has been widely studied, was assessed [9]. Multiple studies report that the microbiota profile changes between different niches along the fish gut, resulting in significant variation in gut functionality [10,11,12]. For example, the pH, which is a fundamental property of the intestinal tract (enabling food digestion and absorption), changes along the gut due to the microbiota activity [13].

The fish gut is a long digestive organ including a foregut, midgut and hindgut, in which the foregut includes the esophagus, stomach and pyloric caeca and the hindgut consists of the distal gut and anus [14]. Although there is no clear boundary to define these gut regions, they generally have different digestion roles, including absorption (i.e., for proteins from simple digestion in the stomach) and digestion (i.e., for the more complex proteins). As an example, the pyloric caeca is where most of the absorption occurs, facilitated by the large surface area, whereas digestion occurs primarily in the midgut [14]. The different roles also correlate with different microbiota niches. The density, composition and function of the microbiota changes along the fish gut [15,16]; especially, there is distinction and variance of the core microbiota and the transient microbiota, which are related to the digesta [17,18]. A recent study has shown that the fish hindgut microbiota has a closer resemblance to the mammalian gut microbiota than to the surrounding environmental microbiota [19], indicating that the core microbiota in the distal gut is not affected by the environment and implies a role in the interaction with the host. As evident from our previous study [9], the composition of the distal gut microbiota correlates with the flesh color. The other end of the gut, the pyloric caeca, has been studied mainly in the context of the dietary components, which affect the gut microbiota [13,20]. Although the change in microbiota composition along the gut has been confirmed [10,11,12], there are no studies reporting the role of the pyloric caeca microbiota in the host phenotype. Interestingly, the pyloric caeca is a more favorable niche for the probionts (i.e., probiotic bacteria) than any other part of the intestine in the Atlantic salmon and Atlantic cod [21], which makes it an important gut region for further investigation.

Given the significance of these two gut regions and the high likelihood of differential microbiota presence between them, both were investigated to obtain a more comprehensive picture of any microbiota correlation with flesh color variation, which might give further insights into the occurrence of commercially unwanted phenotypes (i.e., general paleness or pronounced banding). To address this aim and reduce confounding factors, one sampling event was chosen, at a time-point when flesh color recovery from thermal stress is observed to be differential across individuals from a single population kept under the same conditions and fed the same diet. In this study, the association of the microbiota profile in the pyloric caeca with the flesh color index was examined, and the microbiota profiles in the pyloric caeca and the distal gut and the flesh color were correlated.

## 2. Materials and Methods

### 2.1. Study Design and Sample Collection

The Atlantic salmon individuals analyzed in the current study were sampled on April 2019 from an aquaculture lease in northern Tasmania (Australia) (the same lease as in [9]). The sampling time point was chosen based on commercial historical data showing that flesh color deterioration starts to be manifested in March after the summer thermal stress and a period of poor feed utilization and starts recovering in most of the stock in the following months. The fish were previously stocked in 2018 and maintained in a cage containing approximately 35,000 individuals [9]. When the fish reached 23 months old, fish from two cages, which had similar sizes, were sampled. There was no specific treatment or nutritional trials associated with these fish. When sampled in April, the fish were on a commercial diet containing 40 ppm astaxanthin/40 ppm canthaxanthin. The sampling was conducted over two continuous days in April 2019, and it was carried out before feeding.

A total of 96 individuals were chosen to cover the variation in flesh color and banding status (i.e., differential color tone between the dorsal and the back area of the fillet), resulting in 96 samples each of pyloric caeca and distal gut. The flesh color measurement was carried out visually on two areas of the fillet (front-dorsal and back-central) and based on the Roche SalmoFan Lineal Card (Hoffman-La Roche, Basel, Switzerland) (Figure 1), ranging from 20 to 34, with an assumed minimum marketable grade of 25. When present, the severity of the banding was calculated as the color score difference between the two areas of the fillet assessed and categorized as None (=0), Moderate (=1) or Severe (≥2). The sample details can be found in the Appendix A. Digestive tract samples were collected as described in [9]. Briefly, the fish abdomen was incised, and the pyloric caeca and distal part of the intestine (Figure 2) were aseptically exposed and severed. After opening the intestine longitudinally, a sample was collected (0.5 cm^3^) and then washed three times in sterile 0.9% saline solution to remove non-adherent (allochthonous) bacteria and digesta. The samples were immediately fixed in RNAlater and stored at −80 °C until extraction. Two samples from the rearing water were also collected as the negative control for the background microbiota. Another two blank water samples were also included in the DNA extraction process as the negative control for this process. All procedures were carried out with the approval of the University of the Sunshine Coast Animal Ethics Committee (AN/E/16/12).

### 2.2. DNA Extraction and MiSeq Sequencing

A total of 196 samples were subjected to DNA extraction using the QIAamp BiOstic Bacteremia DNA Kit (Qiagen). DNA quality and quantity were checked with a NanoDrop^TM^ 2000 Spectrophotometer and gel electrophoresis. The DNA was sent to the Ramaciotti Centre for Genomics (University of New South Wales, Sydney, Australia) for the PCR amplification of the hypervariable V3-V4 region between the 341 and 806 nucleotides of the 16S rRNA gene with the specific primers FW-5′-CCTAYGGGRBGCASCAG and RV-5′-GGACTACNNGGGTATCTAAT, and the purified amplicons served for library preparation followed by Illumina sequencing using MiSeq paired-end sequencing.

### 2.3. Microbial Community Profiling

The microbial community profiling analysis was performed using Quantitative Insights into Microbial Ecology (QIIME 2) [22]. The quality filtering was performed using QIIME2 with a Phred score of 19 [22], followed by denoising by the Deblur workflow [23]. Taxonomy was assigned by the VSEARCH consensus taxonomy classifier trained on the GreenGenes database 18_5 with 97% Operational Taxonomic Units (OTUs) [24], followed by the generation of the OTU table.

The OTU table was rarefied at 300 reads per sample, and then, the samples were grouped into categories. There were two datasets: distal gut microbiota data and pyloric caeca microbiota data. To analyze the data, the samples in each dataset were grouped into two categories: flesh color and banding status. In the category flesh color, the data were grouped into four subgroups based on their color assignment: Flesh21-22 (*n* = 8), Flesh23-24 (*n* = 20), Flesh25-26 (*n* = 48) and Flesh27-29 (*n* = 20). In the banding status category, the data were split into three subgroups included None-banding (*n* = 10), Moderate-banding (*n* = 62) and Severe-banding (*n* = 24). The α-diversity and β-diversity were computed using the nonparametric Kruskal–Wallis test and UniFrac distance, respectively [25]. Principal coordinate analysis (PCoA) was performed using Emperor in QIIME 2 [26]. The β-diversity’s multivariate statistical analysis was carried out in QIIME 2 using the Permutational multivariate analysis of variance (PERMANOVA) test [27].

The differential microbial taxa for each group were identified using Linear discriminant analysis Effect Size [28,29], provided by Dr. Huttenhower’s lab, with a Linear discriminant analysis (LDA) effect size of 3.5 and *p*-value < 0.05 (https://huttenhower.sph.harvard.edu/galaxy/). To identify niche taxa that possibly influenced biologically relevant features in response to a change in environmental conditions, the OTU was subjected to QIIME2 Gneiss [30]. Gradient clustering hierarchies were constructed for each category—flesh color and banding status—followed by the construction of the Simplicial Ordinary Least Squares Regression models to obtain the balance trees of the microbiota taxa for each category [30]. Pearson’s correlation was also determined to measure the strength of the correlation between microbiota taxa and a certain phenotype.

### 2.4. Functional and Metabolic Pathway Prediction

The functional annotation of the taxon composition was predicted by PICRUSt [31]. The predictive metagenome was categorized by function, followed by redundancy analysis (RDA) (*p* < 0.05) and Linear discriminant analysis effect size (LEfSE) analysis integrated on a multi-functional web-server, Calypso (LDA 3.5 and *p* < 0.05) [32].

## 3. Results

### 3.1. The Distinct Difference between Microbiota in the Distal Gut and the Pyloric Caeca

After quality control at a Phred score of 19, there was a total of 2,992,577 retained reads, which accounted for 95.3% of the total raw reads. The two datasets of microbiota, those from the distal gut and the pyloric caeca, were pooled together to assess the quality and diversity (α-diversity (Shannon index) and β-diversity, respectively; Figure 3). The Shannon index indicated that the microbiota in the pyloric caeca was richer and more diverse than that of the distal gut (Figure 3A), with significantly differential microbiota composition between these two gut regions (Figure 3B; PERMANOVA test *p* = 0.001). Hence, there are two distinct microbiota populations inhabiting the distal gut and the pyloric caeca in *S. salar*.

Further analysis of the microbiota population in both the distal gut and pyloric caeca indicated that the majority of taxa occupying the distal gut belonged to the class of *Gammaproteobacteria* (Figure 4A, purple circles). However, in the pyloric caeca, almost all the bacteria classes had good presence (Figure 4A, blue circles) with the exception of a significant part of the *Gammaproteobacteria* class. This observation is aligned with the α-diversity (Shannon index) and β-diversity (PCoA) described above (Figure 3). The results of the LEfSe analysis also align with these results by indicating that the representative taxa of the distal gut microbiota were *Vibrionaceae* (F), *Pseudoalteromonadaceae* (F) and *Photobacterium* (G), which belong to only one class, *Gammaproteobacteria*, whereas the representative taxa for the pyloric caeca microbiota were more phylogenetically diverse, with 20 taxa belonging to five classes: *Actinobacteria*, *Betaproteobacteria*, *Alphaproteobacteria*, *Gammaproteobacteria* and *Bacilli* (Figure 4B).

### 3.2. Distal Gut Microbiota Correlates with Salmon Flesh Color

The microbiota in the distal gut dataset, once separated from the pyloric caeca dataset, was analyzed in two categories: flesh color and banding status. In each category, the phenotypes of the subgroups were distanced by the weighted UniFrac distance algorithm, and the statistical difference was analyzed with the PERMANOVA test. However, only the microbiota in the flesh color category was significantly different between phenotypes, with *p* = 0.031, whereas the banding category did not show any correlation with the distal gut microbiota (*p* = 0.143). The distal gut microbiota was then further analyzed by Simplicial Ordinary Least Squares Regression to identify the niches of the microbiota taxa for the flesh color phenotypes. This regression model shows that the flesh color phenotype alone accounted for up to 21.9% of the variance in the entire distal gut microbiota (R^2^ differences = 0.2194). Subsequently, the distal gut microbiota was subjected to Pearson’s correlation analysis to show the related microbiota taxa that were strongly connected to the specific phenotype that formed different patterns (Figure 5). In Figure 4, the low color index Flesh21-22 showed a high correlation with *Ralstonia* (G), *Rhodospirillaceae* (F), *Enterobacteriaceae* (F), *Phyllobacteriaceae* (F), *Microbacteriaceae* (F), *Vibrio* (G) and *Burkholderiales* (O). Flesh23-24 are abundant in *Propionibacterium acnes*, *Photobacterium* (G), *Burkholderia* (G), *Delftia* (G), *Microbacteriaceae* (F) and *Pseudomonas* (G) (Figure 5). The phenotype of Flesh25-26’s pattern is closest to Flesh23-24’s and is highly correlated with *Propionibacterium acnes*, *Photobacterium* (G), *Chryseobacterium* (G), *Vibrio* (G) and *Carnobacterium* (G) (Figure 5). Finally, the highest color index is rich in a distinct group of microbiota, including *Sphingobium* (G), *Vibrionaceae* (F), *Stenotrophomonas* (G), *Bacillus* (G), *Lawsonia* (G), *Xanthomonadaceae* (F), *Methylobacterium* (G) and *Delftia* (G) (Figure 5).

### 3.3. The Correlation of Microbiota in the Pyloric Caeca with Banding Status

Due to the richness and phylogenetic diversity of the pyloric caeca data being higher than those that belonged to the distal gut, the pyloric caeca data were further rarefied with 853 reads per sample. The microbiota in the pyloric caeca dataset was analyzed based on two categories—flesh color and banding status—with the weighted UniFrac distance algorithm, and statistical differences were tested with the PERMANOVA test. The microbiota in this gut region were significantly correlated with both flesh color (PERMANOVA, *p* = 0.021, Figure 6A) and banding status (PERMANOVA, *p* = 0.02, Figure 7A). Subsequently, the microbiota was subjected to a Pearson’s correlation test to identify related microbiota taxa that strongly correlated with a certain phenotype of flesh color (Figure 6B) and banding status (Figure 7B). According to the Pearson’s correlation, Flesh27-29 was related to *Methylobacterium* (G), *Xanthomonadaceae* (F), *Bacillaceae* (F), *Propionibacterium acnes*, *Rhodospirillaceae* (F), *Bradyrhizobiaceae* (F), *Burkholderiales* (O), *Legionella* (F), *Lawsionia* (G), *Pseudomonas* (G), *Brevundimonas diminuta*, *Leucobacter* (G), *Sphingobium* (G), Delftia (G), Ralstonia (G), *Enterobacteriaceae* (F) and *Carnobacterium* (G) (Figure 6B). Meanwhile, Flesh21-22 was related to *Vibrionales* (O), *Nesterenkonia* (G), *Enterobacteriaceae* (F), *Mitochondria* (F), *Enterococcaceae* (F) and *Pseudoalteromonadaceae* (F). However, performing a further statistical step in the LEfSE and sPLS analyses showed that the taxa contributing to Flesh27-29 were only *Carnobacterium* (G) (significantly differential, ANOVA *p* = 0.037) and *Enterobacteriaceae* (F) (ANOVA *p* = 0.29), and the taxa contributing to Flesh23-24 were only *Photobacterium* (F) (ANOVA *p* = 0.13) (Figure 6C,D). The other taxa appearing to be correlated according to Pearson’s correlation neither contributed to nor were differentiated in the flesh color phenotypes.

In the correlation of the pyloric caeca and banding status, None-banding was correlated with *Comamonadaceae* (F), *Rhodospirillaceae* (F), *Nesterenkonia* (G), *Bradyrhizobiaceae* (F), *Ralstonia* (G), *Carnobacterium* (G), *Legionella* (G), *Leucobacter* (G), *Methylobacterium* (G), *Brevundimonas diminuta*, *Enterobacteriaceae* (F), *Pseudomonas* (G), *Delftia* (G), *Burkholderia* (G), *Bacillus* (G), *Sphingobium* (G), *Microbacteriaceae* (F) and *Propionibacterium acnes* (Figure 7B). Meanwhile, Severe-banding was strongly correlated with *Vibrio* (G), *Streptophyta* (O), *Xanthomonadaceae* (F), *Acinetobacter* (G), *Bradyzhizobiaceae* (F), *Truepera* (G), *Geodermatophilaceae* (F), *Enterobacteriaceae* (F), *Pseudomonas* (G), *Enterobacteriaceae* (F), *Burkholderiales* (O) and *Propionibacterium acnes* (Figure 7B). However, in the LEfSE and sPLS analyses, the taxa contributing to the phenotype of None-banding were only *Carnobacterium* (G) (significantly differential, ANOVA *p* = 0.0077) (Figure 7C, D). Only *Acinetobacter* (G), *Enterobacteriaceae* (F) and *Pseudomonas* (G) were significantly differentiated in the severe-banding group (LEfSE analysis, Figure 7C); however, none of them contributed to the phenotype Severe-banding (sPLS analysis, Figure 7D).

Since *Carnobacterium* (G) was the typical taxon in the good color index Flesh27-29 and None-banding, *Carnobacterium* (G) was subjected to a regression model to identify the complex associations between the environmental variables (i.e., color phenotypes) and the microbial taxon. The regression models show that, in the category of flesh color, *Carnobacterium* (G) were significantly correlated with the change in flesh color (Pearson correlation R = 0.401, *p* = 0.0057; significantly differential for Flesh27-29 with ANOVA *p* = 0.037, Figure 8A). Additionally, in the category of banding status, *Carnobacterium* (G) were significantly correlated with the change in banding status (Pearson correlation R = 0.353, *p* = 0.016; significantly differential for None-banding with ANOVA *p* = 0.0077, Figure 8B).

### 3.4. Predictive Function Annotation of Taxon Composition Related to Banding in the Pyloric Caeca

The taxon composition of pyloric caeca was further analyzed to predict the function categories that related to the differential taxa in the banding phenotypes. There were 6909 Kyoto Encyclopedia of Genes and Genomes KEGG groups, predicted and categorized into 328 KEGG metabolism pathways. The predictive functions were significantly distinctive for the banding category (RDA *p* = 0.044, Figure 9A). The functions of taxa differentiated in the None-banding category focused on amino acids, fatty acids, energy (pyruvate), and carotenoid (terpenoid backbone) metabolism (Figure 9B). However, the functions of the taxa in the Severe-banding category focused on sugar metabolism, bacterial chemotaxis, bacterial secretion systems, transcription and translation structure molecules, and peptidase (Figure 9B).

## 4. Discussion

In this study, the effect of microbiota in different niches of the salmon gut on flesh color and banding status was investigated using samples from two different gut regions: the pyloric caeca and distal gut. The microbiota populations from the two regions were demonstrated to be distinct from each other, in agreement with other studies demonstrating that microbiota along the gut niches are different [10,11,12]. 

In a previous study, the correlation between the distal gut microbiota and flesh color was demonstrated [9]. This effect was found to occur independently of the water temperature or the season’s change. In this study, the influence of the distal gut microbiota on the change in flesh color was confirmed. With the sampling in this study taking place at one time point only, the distal gut microbiota still showed variations that correlated with the change in flesh color when using the PERMANOVA test *p* = 0.031 (Figure 5). According to the simplicial ordinary least squares regression model algorithm, the distal gut microbiota composition could explain up to 21.9% of the change in salmon flesh color. In microbiome studies, the regression model of the rooted balance tree is considered significant when the fitting coefficient R^2^ > 0.1 [30,33]. Therefore, this result strengthens the idea of a correlation between the distal gut microbiota and the flesh color. 

In our previous study [9], a low color index was correlated with *Pseudoalteromonadaceae* (F), *Vibrionaceae* (F) and *Enterobacteriaceae* (F) in the distal gut, which also dominated in the low color group in both gut regions - the distal gut and pyloric caeca - in this study (Figure 5 and Figure 6). Despite the different seasons and environmental conditions between the two studies (the two sampling events took place 2 years apart), *Pseudoalteromonadaceae* (F), *Vibrionaceae* (F) and *Enterobacteriaceae* (F) still persisted in the low color index group, indicating their core position as the symbionts in the altered-symbiosis microbiota composition. Altered symbiosis is defined as a status where the microbiota composition changes due to a change in host and environmental stresses; however, this change does not affect the host health [34]. *Pseudoalteromonadaceae* (F), *Vibrionaceae* (F) and *Enterobacteriaceae* (F) belong to *Gammaproteobacteria* (O), which are aerobic bacteria with diverse catabolism pathways that may affect the host physiology [35,36] and, in this specific case, possibly impact flesh color. 

In the high color index group found in our previous sampling event in 2017 [9], the microbiota taxa that correlated with the flesh color were *Xanthomonadaceae* (F), *Bacillaceae* (F), *Mycoplasmataceae* (F), *Phyllobacteriaceae* (F) and *Commamonadaceae* (F). In the current study, *Xanthomonadaceae* (F) and *Bacillaceae* (F) only are in agreement with what was previously found (Figure 5 and Figure 6). These findings imply the core symbiont role of *Xanthomonadaceae* (F) and *Bacillaceae* (F) in the high color index group as being consistent despite the sampling occurring in a different year with different environmental conditions. It also confirms the consistency of the core microbiota within the gut regardless of the change in microbiota composition in the surrounding water and in the diet [7,10,37,38]. *Bacillaceae* (F) is considered as a beneficial bacterial group that is able to support the host’s health by providing antibiotics, vitamins and digestive enzymes [39,40], as well as controlling opportunistic pathogens [41]. *Bacillaceae* (F) has been known as a probiotic solution at the early larval stage of the common snook *Centropomus undecimalis* [42], gilthead sea bream *Sparus aurata* [43] and grouper *Epinephelus coioides* [44]. Furthermore, *Bacillaceae* (F) possesses the carotenoid biosynthesis pathway, and the pigment ranges from yellow to orange and red [45,46]. Therefore, it is possible that *Bacillaceae* (F) naturally contributes to high-quality flesh. On the other hand, *Xanthomonadaceae* (F) was the dominant taxon in the carotenoid metabolism pathway in our pervious study [9]. *Xanthomonadaceae* was firstly found in plants and is able to produce the pigments xanthomonadins [47,48]. *Xanthomonadaceae* (F) is found as a dominant group of bacteria that suppresses pathogenic symbionts in plants [49] and establishes a symbiotic relationship with coral [50,51,52] and the carotenogenic chlorophyte alga *Haematococcus lacustris*, which is able to produce keta-carotenoid astaxanthin [53]. Therefore, in salmon, *Xanthomonadaceae* (F) is possibly a symbiont contributing to the carotenoid biosynthesis pathway. Hence, regardless of the change in the microbiota composition in the high color index group, when comparing the sampling events in 2017 and 2019, and regardless of the difference in gut position, there were two consistent taxa, *Xanthomonadaceae* (F) and *Bacillaceae* (F), that correlated with the strong pink–red flesh color.

The pyloric caeca microbiota data were richer and more diverse than those of the distal gut (Figure 3 and Figure 4). This microbiota was also significantly different between phenotypes—both flesh color and banding categories - according to the weighted UniFrac distance (PERMANOVA *p* = 0.021 and *p* = 0.02) (Figure 6A and Figure 7A). In the high flesh color index group—besides *Xanthomonadaceae* (F) and *Bacillaceae* (F), which were consistent with those in the distal gut microbiota, confirming their core role in most of the digestive tract—the pyloric caeca microbiota also included some new taxa that differentially occupy this gut region (Figure 6 and Figure 7), especially *Carnobacterium* (G), which is highly correlated with both Flesh27-29 and None-banding (Figure 8). *Carnobacterium* (G) belongs to the order of *Lactobacillales*, which is a functional order producing lactic acid as an essential metabolic end-product of carbohydrate fermentation [54]. The effect of lactic acid bacteria (LAB) has been widely studied in Atlantic salmon, since they contribute to maintaining host health [55], stimulate the host immune response and help to protect the host against diseases [56]. The importance of LAB in aquaculture has been extensively reviewed [57,58]. Feeding a LAB-supplemented diet can control the colonization of the beneficial bacteria, leading to better control of diseases and stresses, and enhancing growth in salmonids [59,60,61,62]. The group of *Carnobacterium* has also been known as beneficial bacteria dominantly colonizing various regions of the salmon gut (foregut, midgut and hindgut) and being able to inhibit the growth of pathogens [60]. A previous study has demonstrated that feeding *Carnobacterium* (G) as a probiotic to Atlantic salmon and rainbow trout had no harmful effects and also helped to increase the survival rate during infection challenges [63] due to its ability to produce bacteriocins and antimicrobial compounds [64,65]. Besides the immune-system-supporting function, the role of *Carnobacterium* (G) in fish may have another potential impact: pigmentation. In 2012, Hagi et al. demonstrated that multi-stressors in LAB such as oxidative stress, high temperature stress or envelope stress can be tolerated by carotenoid production [66]. In Hagi’s study, a novel gene for carotenoid production, *crtM*-*crtN*, from *Enterococcus gilvus* was cloned into *Lactococcus lactis*, a LAB, to support this hypothesis [66]. In *Carnobacterium* (G), the carotenoid gene cluster consists of 4,4′-diaponeurosporenoate glycosyltransferase *crtQ*, 15-cis-phytoene synthase *crtM*, phytoene desaturase *crtN,* 4,4′-diaponeurosporene oxidase *crtP* and glycosyl-4,4′-diaponeurosporenoate acyltransferase *crtO* [67]. Hence, it is possible that *Carnobacterium* (G) can evoke their carotenoid synthesis cascade as a response to tolerate their stresses. *Carnobacterium* (G) were found in the Flesh27-29 and None-banding fish. It is possible that *Carnobacterium* (G) originated carotenoid is one of the carotenoid sources for the host. Additionally, *Carnobacterium* (G), with their ability of immune support, might be in part responsible for the host’s better health in response to the thermal stress and therefore leads to the more efficient feed assimilation. Hence, feeding *Carnobacterium* (G) to Atlantic salmon as a probiotic to improve health, the immune system and pigment deposition is a potential biotechnology application. However, it is still not clear whether the correlations observed in the microbiota are the cause or the effect of the phenotypes observed. Therefore, further investigation of *Carnobacterium* (G) and their carotenoid gene expression under multiple stresses is needed. 

The pyloric caeca data also showed a correlation between Severe-banding and a group of microbiota that were significantly differentiated: *Acinetobacter* (G), *Enterobacteriaceae* (F) and *Pseudomonas* (G) (Figure 7C). *Acinetobacter* (G) is known as a typical opportunistic pathogen that causes Acinetobacter disease in Atlantic salmon, channel catfish, common carp, rainbow trout and striped bass [68]. *Pseudomonas* (G) is found as an acute septicemic bacterial disease in fish [68,69,70]. More importantly, *Acinetobacter* (G) and *Pseudomonas* (G) are known as multi-antibiotic-resistant pathogens in aquaculture [71,72,73,74], and *Acinetobacter* (G) has been used as an indicator of antibiotic resistance in the aquatic environment [75,76,77]. The antibiotic resistance of *Acinetobacter* (G) is of concern since its antibiotic resistance gene can be transferred horizontally to the surrounding microbiota community [78]. All the fish sampled in this study appeared healthy; however, those that were dominated by *Acinetobacter* (G) and other opportunistic bacteria might be experiencing an altered-symbiosis status. In an altered-symbiosis stage, although the health appearance does not change, the host initiates an immune response in an attempt to overcome this short-term stress and return to a normal host–microbiome relationship [34]. Carotenoids are known to have multiple roles, both non-specific and specific, in the immune system [79]. There is evidence that carotenoid-supplemented diets can enhance the immune response parameters in rainbow trout *Oncorhynchus mykiss* [80], common carp *Cyprinus carpio* [81] and Atlantic salmon *Salmo salar* [82]. The carotenoid trade-off for pigmentation and immunity has been demonstrated in the polychromatic midas cichlid *Amphilophus citrinellus* [83]. In the polychromatic midas cichlid, the fish itself must decide to allocate the limited carotenoid source for either pigment deposition or the enhancement of the immune system in response to stress [83]. Therefore, we hypothesize there is a trade-off between pigmentation and the immune response in the usage of carotenoids in Atlantic salmon, which leads to the pale flesh and Severe-banding. However, it is unclear if the carotenoid source for this trade-off is derived from the diet or is withdrawn from the fish muscle. A further experiment to examine the carotenoid level retained in the muscle of the fish in this study has been conducted and will be reported in a separate publication. These hypotheses need to be further investigated with supporting evidence from gene expression studies.

In the function analysis of the taxa, the composition was differentiated in the different banding phenotypes and the metabolism pathways were significantly distinctive (*p* = 0.04, Figure 9A). In the no-banding group, the functions of the microbiota taxa focused on amino acids, fatty acids, energy (i.e., pyruvate) and carotenoid (i.e., terpenoid backbone) metabolism (Figure 9B). In this group, the representing taxa were *Carnobacterium* (G) (Figure 8), which belongs to LAB, strongly supporting the hypothesis above that the beneficial taxa in this symbiosis stage communicated with the host via the intermediates in their metabolism cascade such as amino acids, carotenoids and fatty acids. Additionally, fat and fatty acids play a major role in the regulation of the transporter expression in the intestine, which affects carotenoid absorption [84]. Thus, in this no-banding group, the differential taxa occupy the pyloric caeca in a symbiotic manner. Conversely, the functions of the microbiota taxa in Severe-banding focused on sugar metabolism, bacterial chemotaxis, bacterial secretion systems, transcription and translation structural molecules, and peptidase (Figure 9B). Bacterial chemotaxis is a signal–feedback loop between the bacteria and the environment [85]. This loop exists to maintain the communication and ensure the survival of the bacteria themselves in either favored or stressed environments [85]. Any action performed by the bacteria on the environment returns as a physical-chemical signal that will be processed, regulated and decided by the bacteria before any new action is performed [85]. The other metabolism pathways in the severe-banding group include sugar metabolism, bacterial secretion systems, transcription and translation structural molecules, and peptidase, all helping the bacteria to achieve the chemotaxis [86]. In terms of host infection, it has been demonstrated that bacterial chemotaxis has a key role in the pathogenicity [86]. This once again supports the hypothesis above, that there was an altered symbiosis occurring in the severe-banding group.

## 5. Conclusions

The correlation of the distal gut microbiota and salmon flesh color was re-confirmed in this study. *Xanthomonadaceae* (F) and *Bacillaceae* (F) were the two consistent taxa in individuals displaying high color at the two sampling events (2017 and 2019). Additionally, the distinction between the microbiota in the distal gut and in the pyloric caeca was confirmed. This study has demonstrated the significant correlation of *Carnobacterium* (G) and high color and the absence of regional color loss (i.e., banding). We also demonstrated the correlation of the salmon pyloric caeca microbiota and the banding phenomenon. We hypothesize that there is a trade-off between the coloration and the immune activity in Atlantic salmon. This study further supports the role of microbiota in flesh color dynamics and opens a path for managing this important commercial characteristic through microbiome manipulations.

## Figures and Tables

**Figure 1 microorganisms-08-01244-f001:**
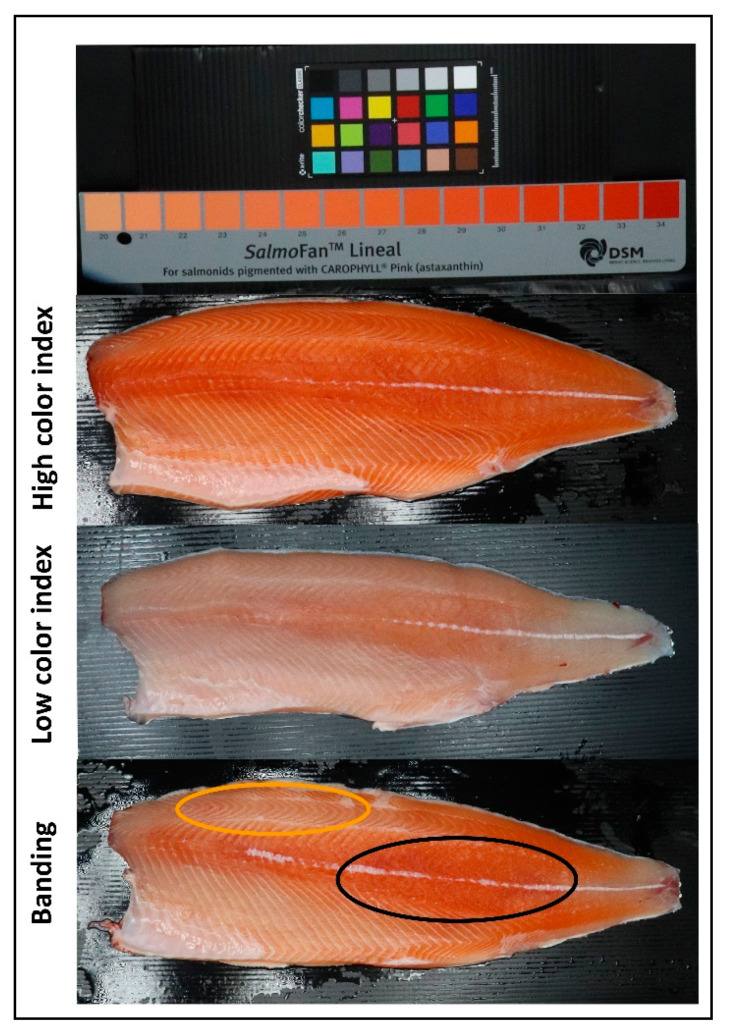
Examples of salmon fillets including high color index with pink–red color over whole fillet, low color index with pale color over whole fillet and banding with a color difference between the dorsal (orange oval) and central back areas (black oval).

**Figure 2 microorganisms-08-01244-f002:**
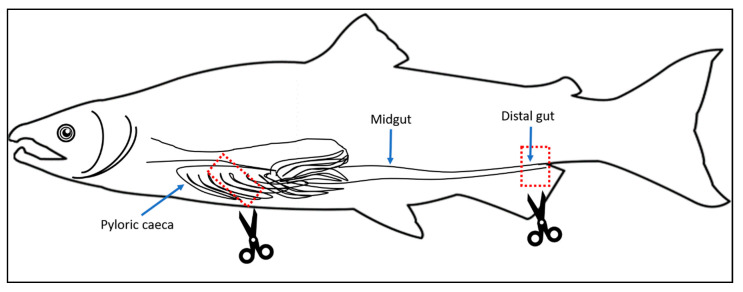
Schematic drawing of the salmon gastrointestinal tract with pyloric caeca: pyloric caeca, midgut and distal gut. Dashed red rectangles and scissors denote the two regions collected in this study.

**Figure 3 microorganisms-08-01244-f003:**
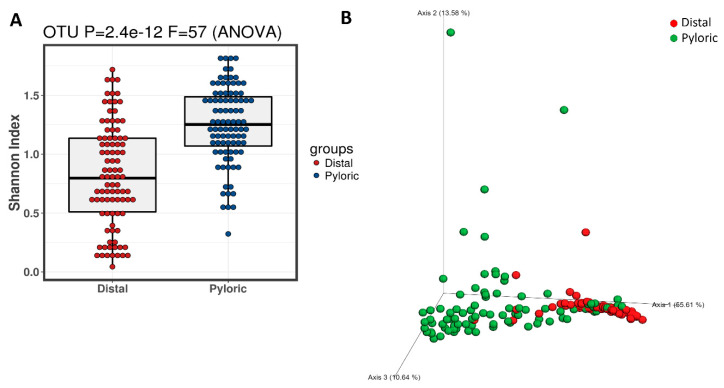
(**A**) The α-diversity (Shannon index) and (**B**) β-diversity (principal coordinate analysis (PCoA)) by the weighted UniFrac distance between the microbiota in the distal gut and the microbiota in the pyloric caeca.

**Figure 4 microorganisms-08-01244-f004:**
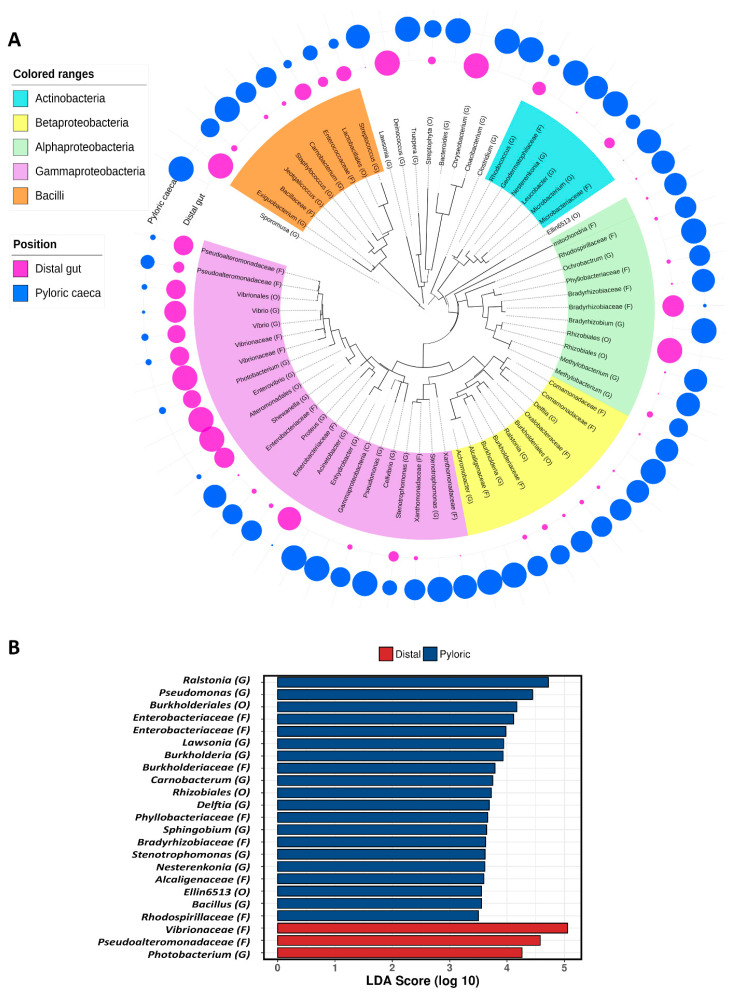
(**A**) A phylogenetic tree representing the microbiota of both distal gut and pyloric caeca. The bacteria classes of *Actinobacteria*, *Betaproteobacteria*, *Alphaproteobacteria*, *Gammaproteobacteria* and Bacilli are highlighted in sky-blue, yellow, light green, pink and orange, respectively. The outer rings of circles indicate the presence, as percentages, of the corresponding taxa in the distal gut (purple circles) and the pyloric caeca (blue circles). (**B**) The differential taxa representing the distal gut (red bars) and pyloric caeca (blue bars). Phylum, class, order, family, genus and species are shortened to P, C, O, F, G and S, respectively.

**Figure 5 microorganisms-08-01244-f005:**
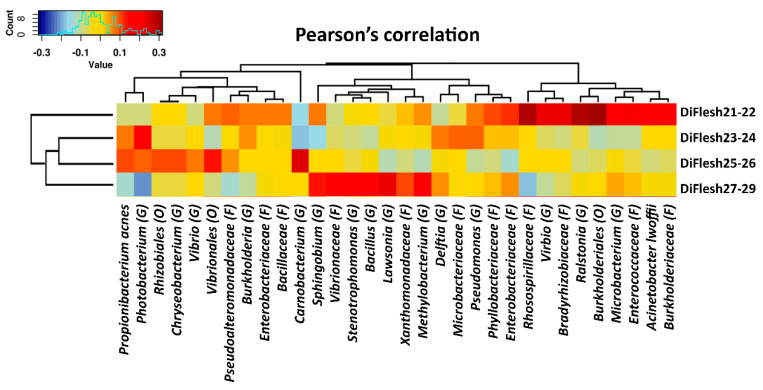
Pearson’s correlation between the distal gut microbiota and flesh color phenotypes. The flesh color phenotypes are presented as Flesh21-22 to Flesh27-29. The prefix “Di” represents the “distal gut” dataset. Phylum, class, order, family, genus and species are shortened to P, C, O, F, G and S, respectively.

**Figure 6 microorganisms-08-01244-f006:**
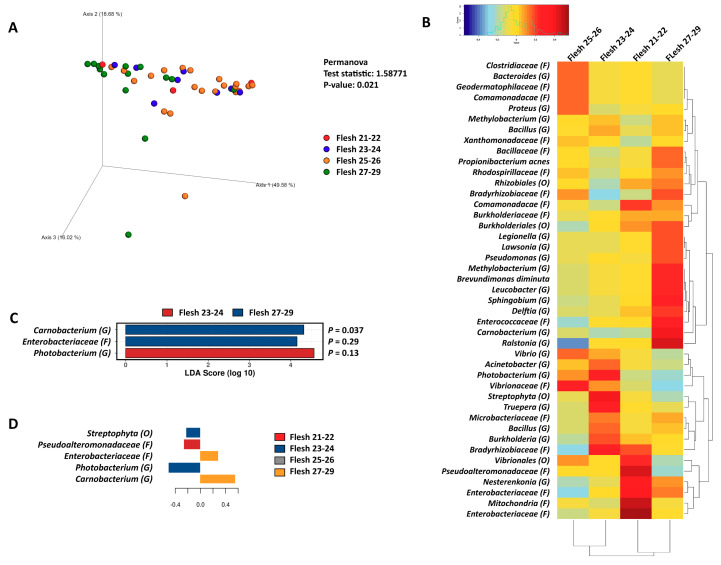
(**A**) Principal coordinate analysis (PCoA) of the pyloric caeca microbiota, which was distanced with the weighted UniFrac algorithm with the PERMANOVA test. The flesh color phenotypes are presented as follows: Flesh21-22 in red, Flesh23-24 in blue, Flesh25-26 in orange and Flesh27-29 in green. (**B**) Pearson’s correlation of the pyloric caeca microbiota and flesh color phenotypes. The flesh color phenotypes are presented as Flesh21-22 to Flesh27-29. (**C**) LEfSE analysis with ANOVA test for each differential taxon; Flesh23-24 in red and Flesh27-29 in blue. (**D**) Spare partial least squares regression analysis to reveal the composition of the taxa that contributed to the phenotypes: Flesh21-22 in red, Flesh23-24 in blue, Flesh25-26 in grey and Flesh27-29 in yellow. Phylum, class, order, family, genus and species are shortened to P, C, O, F, G and S, respectively.

**Figure 7 microorganisms-08-01244-f007:**
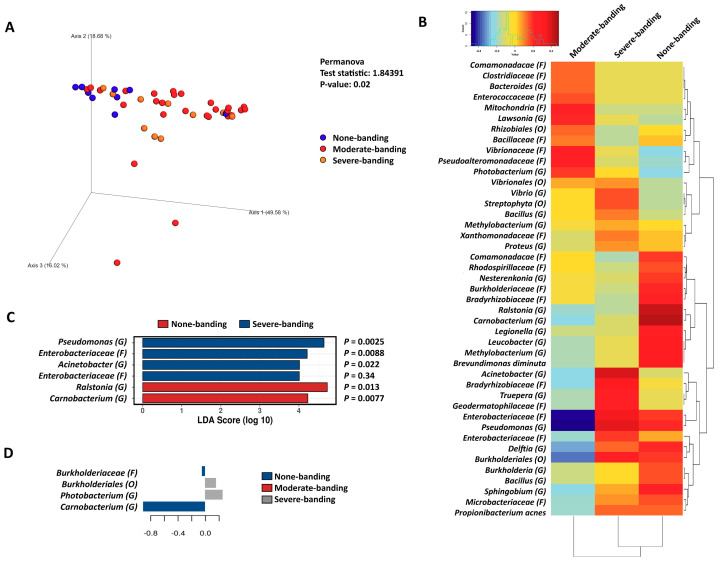
(**A**) Principal coordinate analysis (PCoA) of the pyloric caeca microbiota, which was distanced with the weighted UniFrac algorithm with the PERMANOVA test. The banding phenotypes are presented as follows: None-banding in blue, Moderate-banding in red, and Severe-banding in orange. (**B**) Pearson’s correlation of the pyloric caeca microbiota and banding phenotypes: None-banding, Moderate-banding and Severe-banding. (**C**) LEfSE analysis with ANOVA test for each differential taxon; None-banding in red and Severe-banding in blue. (**D**) Spare partial least squares regression analysis to reveal the composition taxa that contributed to the phenotypes: None-banding in blue, Moderate-banding and Severe-banding in grey. Phylum, class, order, family, genus and species are shortened to P, C, O, F, G and S, respectively.

**Figure 8 microorganisms-08-01244-f008:**
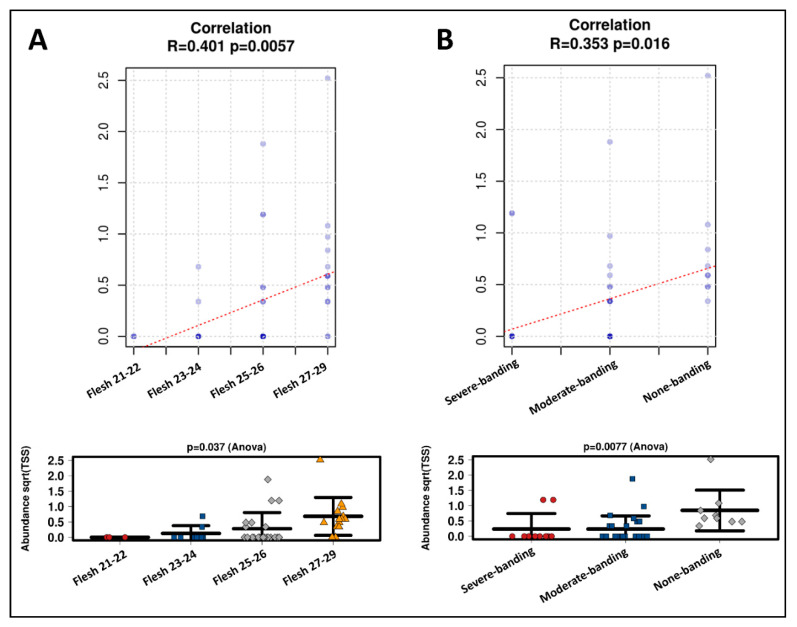
Regression model analysis to identify complex associations between phenotypes—(**A**) flesh-color and (**B**) banding status—and Carnobacterium (G).

**Figure 9 microorganisms-08-01244-f009:**
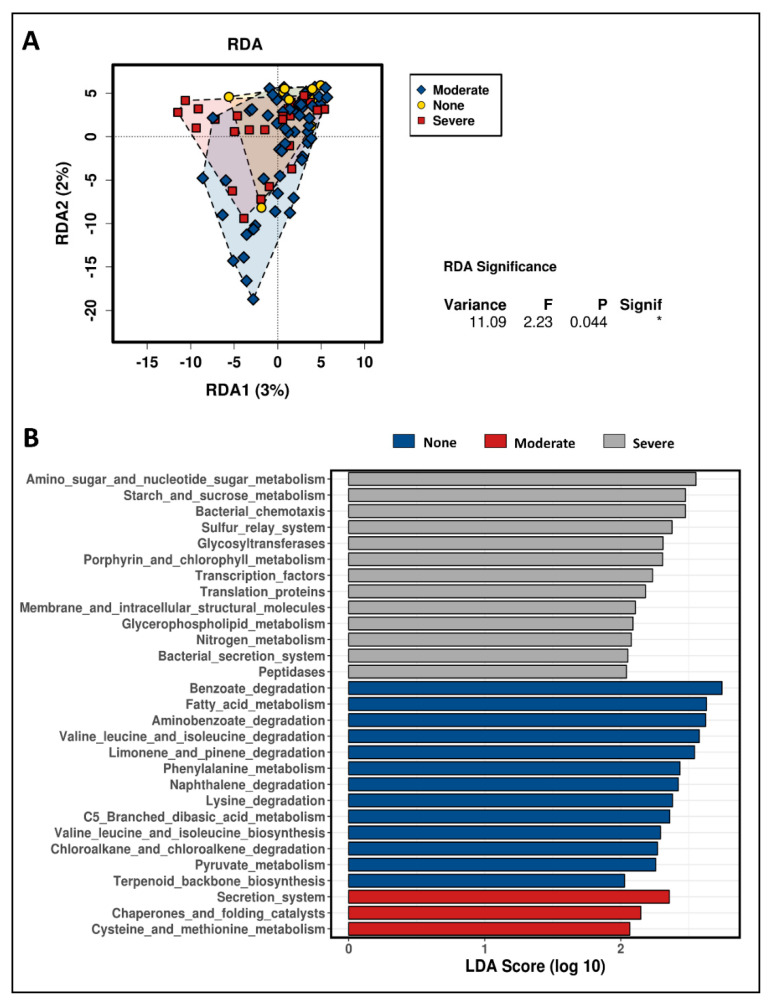
Functional annotation of the pyloric caeca taxon composition differentiated in the different banding phenotypes. (**A**) Redundancy RDA analysis with * *p* < 0.05, and (**B**) LEfSE analysis.

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
