# Peer review of "Assessing the Pyloric Caeca and Distal Gut Microbiota Correlation with Flesh Color in Atlantic Salmon (Salmo salar L., 1758)"

_microorganisms, 2020, doi:10.3390/microorganisms8081244_

Round 1

Reviewer 1 Report

The manuscript evaluated the correlation of intestinal microbial communities with variation in flesh color in Atlantic Salmon using 16S rRNA gene amplicon sequencing. The idea is really interesting, as understanding the microbiota community associated with phenotypic traits is an interesting topic. The manuscript is well written, and the study is well designed, and the author did an in-depth analysis. However, the study presents some issues that need to be addressed prior to publication.

General Comments:

  1. MS has a major problem with reference formatting ((Error! Reference source not found)  making it hard to read.
  2. Did the author check if any of the environmental conditions (PH, temperature, water chemistry, diet, tank) interfered with the results? What could be the source of the variation in the bacteria? diet, feed or something else? Were there any efforts made to sequence water and diet? Because environmental factors have significant impact on determining host-microbial assemblages as well as host phenotype.
  3. I think it’s better to include pictures of fish showing color deterioration in a supplementary file or in the manuscript itself.
  4. What is the expectable amplicon size for V3-V4 hypervariable region?
  5. The author did not mention the number of reads before QC and after QC. How many reads were used for downstream analysis? It should be included in the methods section before downstream analysis.
  6. The description of the functional annotation for different taxa might be interesting. I suggest the author run Tax4Fun for this (R code for Tax4Fun in rainbow trout and Salmon paper should be available).

Specific Comments

  1. L 119-120 I assume the author did PE sequencing, but it might be worth to include it in the manuscript.
  2. L 22-23 Beta diversity measures the differences in microbial assemblages between group of samples, how the author depicts microbial assemblages that correlated to color evenness based on beta diversity without running functional annotation?
  3. L 124-126 Which database was used to align the sequences? (Silvadatabase or Greengenes)
  4. L 164-165 While performing LfSe analysis, the author did not mention any steps to customize the p-values and effect size cutoffs for significant taxa.

Reviewer 2 Report

This paper offers insights into the probable interaction of the host microbiota and flesh quality in salmon. Specifically, they focused the study on pyloric caeca and distal gut. The result provides a significant stride in our understanding of the potential involvement of gut microbiota on host phenotype. This is particularly relevant in salmon farming where manipulation of microbiota is considered a viable strategy to improve health and flesh quality/colour is a significant determinant of market value.

Several issues, enumerated below, must be addressed first before the publication of this manuscript.

L76-77. In what aspect it is favourable?

L90, What’s the size? Is there any relationship between the size and the correlation observed (microbiota vs flesh colour)? It would be interesting to see the distribution of sizes as it may provide insights into the potential contributory factor of this trait.

L91. “One time-point” - Were the samples collected on the same day or over several days during a specific production period? Even though some parameters have been reported in paper 9, it would be beneficial to mention some important information here for contextualisation. F. ex, feeding before sampling, cage, treatments, etc.

L167. Quantify and specify. Diverse such as?

L203. Correlation is a statistical test. “potential correlation” sounds a bit off.

L307: Were there previous studies demonstrating the role of Bacillaceae on pigment production/synthesis?

L313-325. These are excellent studies to support the results of the present study. However, I found that this part was written poorly. This part is like a smorgasbord of information. Should be simplified and the flow of thought must be improved.

L326. Is there any information as to why the pyloric caeca is richer compared with eg the distal gut? Any inference based on the microenvironments?

L359. Carotenoid content of the diet must be provided. Important for reproducibility.

L386-389. This hypothesis will be more robust if the authors provide the carotenoid content of the diet, the muscle, as well as the tissues under study. I highly recommend that this should be done.

L398-399. This hypothesis is too weak to be proposed, given the results of the study. No data have been provided on the immune status of the fish.

Others

There are a lot of “Error! Reference source not found”. This has to be fixed.

Round 2

Reviewer 1 Report

The manuscript overall has been thoroughly improved from previous version. The author addressed all the comments. Here are few minor corrections that may need to be done prior to publication.

  1. It’s better to include figure 1 in methods section (study design and sample collection) rather than introduction.
  2. L168 2.4 I might suggest changing the title to “Functional and metabolic pathways prediction”
  3. L169 replace word function by “functional”
  4. L 204 I suggest author to change topic 2  to “ Distal gut microbiota correlates with salmon flesh color”
  5. L303 Fig. 9 replace word function by “functional”

Author Response

Reviewer 1: The manuscript overall has been thoroughly improved from previous version. The author addressed all the comments. Here are few minor corrections that may need to be done prior to publication.

We thank the reviewer for taking time to consider our work again. Here, we would like to address the reviewer's specific concerns. Please note that with the changes, the line numbers changed, and the new line numbers are provided.

  1. It’s better to include figure 1 in methods section (study design and sample collection) rather than introduction.

Response: Figure 1 has been relocated to method section at L107-110.

  1. L168 2.4 I might suggest changing the title to “Functional and metabolic pathways prediction”

Response: L163, the title of 2.4 section has been changed to “Functional and metabolic pathways prediction”.

  1. L169 replace word function by “functional”

Response: L164, “function” has been replaced by “functional”.

  1. L 204 I suggest author to change topic 2  to “ Distal gut microbiota correlates with salmon flesh color”

Response: L197 The title of 3.2 section has been changed to “Distal gut microbiota correlates with salmon flesh color”.

  1. L303 Fig. 9 replace word function by “functional”

Response: L293 Figure 9, “function” has been replaced by “functional”.

Reviewer 2 Report

The authors have made significant efforts in addressing my concerns in the earlier version of the manuscript. 

The main issue that I have raised earlier is the level of asthaxantin that must be provided to support the foregoing hypothesis. The authors argued that though this aspect has been investigated as well, the result is out of scope of the 1st author's study and will be reported in a separate publication. This is a very important aspect that would have strengtened the hypothesis and implications of the paper. Thus, I find the rebuttal a bit weak and digressive. I will take the words of the authors for now that this will be reported in a separate publication in the future. But this should be mentioned in the final version of the paper for publication. 

Author Response

Reviewer 2:

The authors have made significant efforts in addressing my concerns in the earlier version of the manuscript.

The main issue that I have raised earlier is the level of asthaxantin that must be provided to support the foregoing hypothesis. The authors argued that though this aspect has been investigated as well, the result is out of scope of the 1st author's study and will be reported in a separate publication. This is a very important aspect that would have strengtened the hypothesis and implications of the paper. Thus, I find the rebuttal a bit weak and digressive. I will take the words of the authors for now that this will be reported in a separate publication in the future. But this should be mentioned in the final version of the paper for publication.

We thank the reviewer for taking time to consider our work again.

The astaxanthin/canthaxanthin level in the diet has been provided in the previous revision at L103. A statement has been added in the discussion at L401-403 as “A further experiment to examine the carotenoid level retained in the muscle of the fish in this study has been conducted and will be reported in a separate publication”.